# Strengthening midwifery in the South-East Asian region: A scoping review of midwifery-related research

**Georgia Griffin**[1,2], **Zoe Bradfield**[2], **Kyu Kyu Than**[3], **Rachel Smith**[1], **Ai Tanimizu**[4], **Neena Raina**[4], **Caroline S. E. Homer**[1] *

**1** Maternal, Child and Adolescent Health Program, Burnet Institute, Melbourne, Australia, **2** Faculty of Health Sciences, Curtin University, Perth, Australia, **3** Burnet Myanmar Program, Burnet Institute, Yangon, Myanmar, **4** Family and Gender through the Life Course, South-East Asia Regional Office, World Health Organization, New Delhi, India

* Caroline.Homer@burnet.edu.au

**Data Availability Statement:** All data are provided in the Supplementary File S2 Table.

**Funding:** The WHO South East Asian Regional Office (WHO SEARO) commissioned and funded this review. Two colleagues from the WHO SEARO

## Abstract

Improving sexual, reproductive, maternal, newborn, and adolescent health outcomes necessitates greater commitment to, and investments in, midwifery. To identify future research priorities to advance and strengthen midwifery, we conducted a scoping review to synthesise and report areas of midwifery that have been explored in the previous 10 years in the 11 countries of the World Health Organization's South-East Asia region. Electronic peer-reviewed databases were searched for primary peer-reviewed research published in any language, published between January 2012 and December 2022 inclusive. A total of 7086 citations were screened against the review inclusion criteria. After screening and full text review, 195 sources were included. There were 94 quantitative (48.2%), 67 qualitative (34.4%) and 31 mixed methods (15.9%) studies. The majority were from Indonesia (n = 93, 47.7%), India (n = 41, 21.0%) and Bangladesh (n = 26, 13.3%). There were no sources identified from the Democratic People's Republic of Korea or the Maldives. We mapped the findings against six priority areas adapted from the 2021 State of the World's Midwifery Report and Regional Strategic Directions for Strengthening Midwifery in the South-East Asia region (2020–2024): practice or service delivery (n = 73, 37.4%), pre-service education (n = 60, 30.8%), in-service education or continuing professional development (n = 51, 26.2%), workforce management (n = 46, 23.6%), governance and regulation (n = 21, 10.8%) and leadership (n = 12, 6.2%). Most were published by authors with affiliations from the country where the research was conducted. The volume of published midwifery research reflects country-specific investment in developing a midwifery workforce, and the transition to midwifery-led care. There was variation between countries in how midwife was defined, education pathways, professional regulation, education accreditation, governance models and scope of practice. Further evaluation of the return on investment in midwifery education, regulation, deployment and retention to support strategic decision-making is recommended. Key elements of leadership requiring further exploration included career pathways, education and development needs and regulatory frameworks to support and embed effective midwifery leadership at all levels of health service governance.

office were involved in the study design, interpretation and decision to publish. They are included as co-authors (Ai Tanimizu and Neena Raina). Caroline Homer is also supported by an Australian National Health and Medical Research Council (NHMRC) Investigator Grant (App ID: 2016379).

**Competing interests:** The authors have declared that no competing interests exist.

## Introduction

Midwives can provide care across the spectrum of sexual and reproductive health, from health promotion and education activities to complex, life-saving interventions when practising to the full extent of their scope of practice [1]. Midwives who are regulated, educated and work in enabling environments contribute to lower maternal and neonatal mortality rates, increase access to family planning and reproductive healthcare services, and can enhance gender equality and empowerment in communities [1, 2]. Thus, empowering midwives by prioritising their education and training is crucial to achieving sustainable improvements in sexual, reproductive, maternal, neonatal and adolescent health outcomes [1].

There has been increasing investment in the development and expansion of midwifery as a profession in many regions including in South-East Asia [3]. The 11 countries in World Health Organisation's South-East Asia Regional Office of the (WHO SEARO) (Bangladesh, Bhutan, the Democratic People's Republic (DPR) of Korea, India, Indonesia, Maldives, Myanmar, Nepal, Sri Lanka, Thailand and Timor-Leste) have committed to strengthening their midwifery workforce. This investment in the professionalisation and expansion of midwifery is a critical step towards improving maternal mortality rates along with broader gains in women's sexual and reproductive rights in South-East Asia [4]. Midwifery needs to be integrated into pre-existing health systems to meet the specific needs of each country.

Strategic directions for preparing and strengthening the midwifery workforce are outlined in the Regional Strategic Directions for Strengthening Midwifery in the South-East Asia Region (2020–2024) [3]. Five key elements were outlined in this report: governance and regulation; education and training; workforce planning and management; practice and service delivery; and research and evidence. These key elements echo the 2021 State of the World's Midwifery Report key areas for investment [1]. Workforce regulation and governance including the implementation of legislation, development of professional midwifery associations, and education programmes designed to international standards, are essential components in the establishment of midwifery [5]. Of key importance also, is the identification of, and implementation of strategies to overcome systematic barriers to the integration of high-quality midwifery [6].

Research is needed for the generation of evidence to inform midwifery practice and policy to respond so specific population needs [3]. It is not clear what midwifery research has been conducted in the South-East Asia region. Thus, the overall aim of this scoping review was to identify what areas of midwifery (governance and regulation; education and training; workforce management; practice and service delivery including midwifery models of care; leadership; other relevant topics) have been explored in the last 10 years in 11 countries in South-East Asia and to identify future research priorities to advance and strengthen midwifery.

## Methods

We conducted a scoping review, guided by the approach outlined by Arksey and O'Malley [7] and further defined by Levac and colleagues [8]. Five phases were followed: i) identifying the research question; ii) identifying the relevant literature; iii) study selection; iv) charting the data; and v) collating, summarising and reporting the results.

### Identifying the research question

The research question was identified jointly by the study team and the WHO SEARO team who are supporting the developing of the regional midwifery strategy.

### Identifying the literature

The investigator team developed a search strategy to identify the literature to address the research question. Sources were eligible for inclusion if they reported on governance, regulation, education, training, workforce management, practice, service delivery, midwifery models of care, or leadership, published between January 2012 and December 2022 inclusive. Sources could report on midwifery in one of 11 countries: Bangladesh, Bhutan, DPR Korea, India, Indonesia, Maldives, Myanmar, Nepal, Sri Lanka, Thailand or Timor-Leste. Sources were excluded if they did not report on primary research, such as opinion pieces, discussion papers or systematic reviews, were not peer-reviewed or focused on maternal outcomes or experiences.

For the purpose of this review, midwifery was defined as specifically relating to midwives as defined by the International Confederation of Midwives, and not relating to traditional birth attendants, or unregistered or unregulated practitioners [3]. Auxiliary nurse-midwives and auxiliary midwives were also included, in recognition of the key role these associate midwifery professionals play in the emergence of midwifery in the South-East Asia region [3].

We searched the following databases: CINAHL, Dimensions, Embase, Emcare, Global Health, MEDLINE, Maternity & Infant Care, ProQuest, PsycINFO, Scopus, and Web of Science. Key search terms were also entered into Google Scholar and the first 10 pages of search results were downloaded. Citation tracking was conducted on systematic reviews identified through this search strategy. Finally, grey literature identified by WHO SEARO stakeholders were evaluated against the eligibility criteria for inclusion. S1 Table shows the search terms used. Table 1 presents the results of the MEDLINE search strategy, conducted 18 March 2023.

### Study selection

Sources were imported into Endnote [9]. Duplicates were manually removed. Citations were uploaded into Covidence [10]. Further duplicates were removed. One author independently conducted initial title and abstract screening. Two authors undertook full-text review. Discrepancies were discussed until consensus was achieved.

### Data charting, collating and summarising

The data were extracted into an Excel file that included the details on all the included papers. Quantitative analyses using pivot tables were undertaken to describe the data and synthesize the findings. Study quality was not appraised, as scoping reviews include a broad range of evidence sources and this is not usually possible [11]. Meta-analysis was not undertaken. The scoping review findings are reported according to the Preferred Reporting Items for Systematic reviews and Meta-Analyses extension for Scoping Reviews (PRISMA-ScR) format [12].

### Results

In total, 13,185 citations were identified from the electronic peer-reviewed databases and Google Scholar and uploaded into Covidence [10]. A total 6099 duplicate citations were removed prior to title and abstract screening. The remaining 7086 citations underwent title and abstract screening, and 6462 citations were excluded. Three sources could not be retrieved. A further 430 citations were excluded during full-text review because they were: not about midwives

**Table 1. MEDLINE search strategy.**

| Search | Terms | Records retrieved 18 March 2023 |
|---|---|---|
| 1 | Midwi* OR "nurse midwi*" OR "nurse-midwi*" | 40682 |
| 2 | Midwifery/ | 21197 |
| 3 | Nurse Midwives/ | 7534 |
| 4 | Maternal Health Services/ | 16292 |
| 5 | "Southeast Asia" OR "South East Asia" OR "South-east Asia" OR "South Asia" OR Bangladesh* OR Nepal* OR Bhutan* OR Korea* OR India* OR Indonesia* OR Maldives OR Myanmar OR Burm* OR "Sri Lanka*" OR Thai* OR Timor* | 506384 |
| 6 | Asia, Southeastern/ | 8571 |
| 7 | Bangladesh/ | 14278 |
| 8 | Bhutan/ | 674 |
| 9 | Democratic People's Republic of Korea/ | 295 |
| 10 | India/ | 118433 |
| 11 | Indonesia/ | 13532 |
| 12 | Maldives/ | 7 |
| 13 | Myanmar/ | 3173 |
| 14 | Nepal/ | 10630 |
| 15 | Sri Lanka/ | 7077 |
| 16 | Thailand/ | 30645 |
| 17 | Timor-Leste/ | 271 |
| 18 | 1 OR 2 OR 3 OR 4 | 52950 |
| 19 | 5 OR 6 OR 7 OR 8 OR 9 OR 10 OR 11 OR 12 OR 13 OR 14 OR 15 OR 16 OR 17 | 509912 |
| 20 | 18 AND 19 | 3053 |
| 21 | Limit 20 to yr ="2012–2022" | 1507 |

(n = 139), not primary research (n = 130), focused on patient outcomes (n = 83), reporting on countries outside the region (n = 26), about the COVID-19 pandemic (n = 8), not peer-reviewed (n = 4), or outside the date range (n = 2). An additional four studies were included from citation tracking. A final n = 195 sources were deemed eligible for inclusion. Fig 1 presents a PRISMA flow diagram outlining the study selection process. Included sources are presented in the S2 Table.

The majority reported on midwifery in Indonesia (n = 93, 47.7%), India (n = 41, 21.0%) and Bangladesh (n = 26, 13.3%). Three sources reported on multiple countries in the region. No identified sources reported on midwifery on DPR Korea or the Maldives. The majority of sources were published by authors with affiliations from the country where the research was conducted in (n = 168, 86.2%).

Included sources reported on a wide range of midwives and midwifery stakeholders. There was significant variation in how midwives were titled and defined. Most frequently, the research focus was midwives or nurse midwives (n = 101, 51.8%), auxiliary midwives or auxiliary nurse midwives (n = 33, 16.9%) and students or trainees (n = 33, 16.9%). Auxiliary midwives or auxiliary nurse midwives were reported on in India (n = 21, 10.8%), Myanmar (n = 6, 3.1%) and Nepal (n = 6, 3.1%). Midwifery leaders or educators were also frequently reported on (n = 26, 13.3%) and included clinical facilitators, mentors and policymakers. Public health midwives (n = 9, 4.6%) were reported on solely in Sri Lanka. Finally, 14 sources (7.2%) reported broadly on the midwifery profession or education (Table 2).

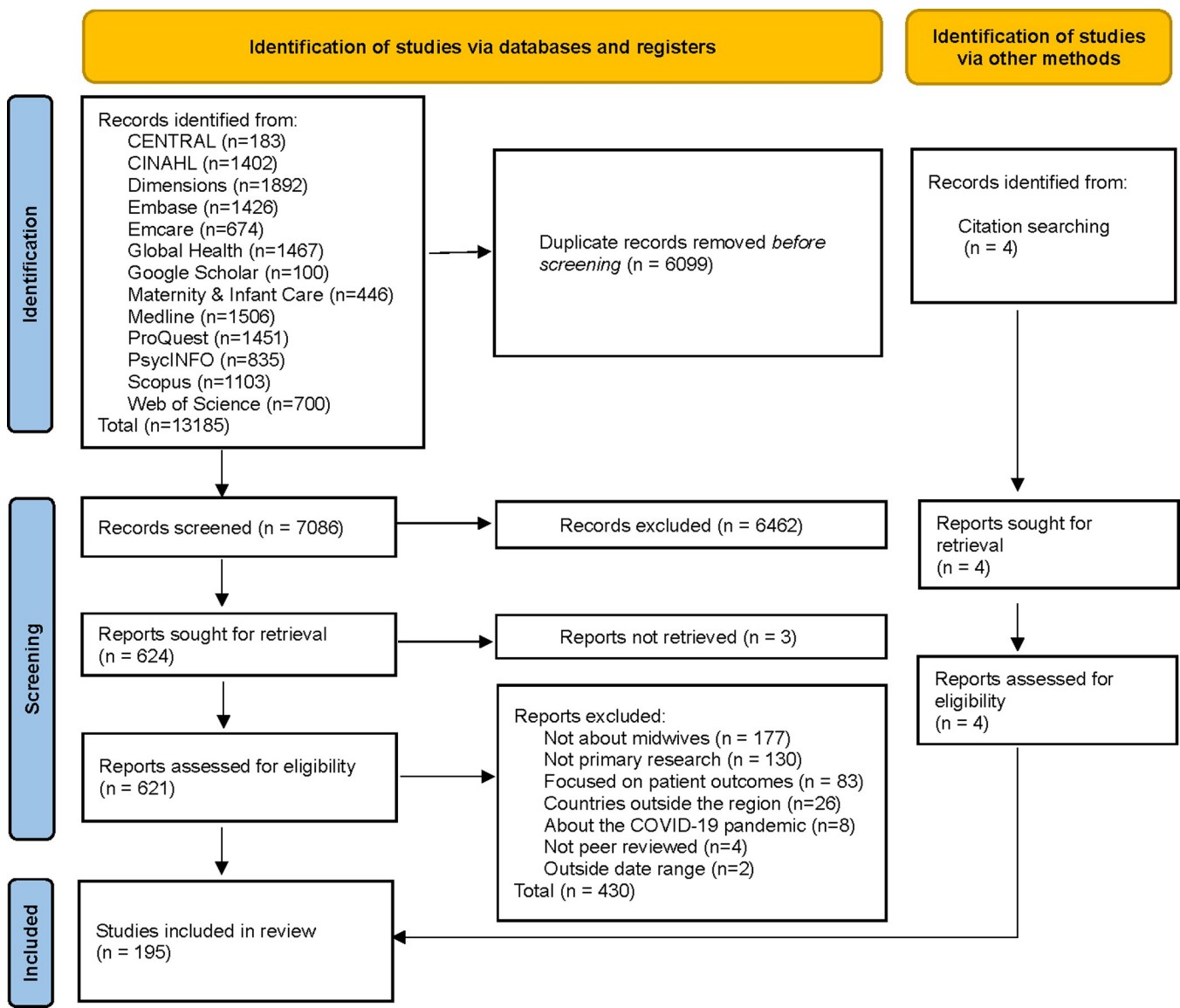

**Fig 1. PRISMA flow diagram.**

The findings were mapped against six focus areas adapted from the Global strategic directions for nursing and midwifery 2021–2025 [1] and the Regional Strategic Directions for Strengthening Midwifery in the South-East Asia Region (2020–2024) [3]. The focus areas were: 1) Practice or service delivery; 2) Pre-service education; 3) In-service education and continuous professional development (CPD); 4) Workforce management; 5) Governance and regulation; and 6) Leadership (Fig 2). Sources could be categorised into more than one focus area.

The most common focus area was related to midwifery practice or service delivery (n = 73, 37.4%). This was followed by pre-service education (n = 60, 30.8%) and in-service education/ CPD (n = 51, 26.2%). The least common focus area was midwifery leadership (n = 12, 6.2%). Fig 2 shows the density of focus areas by country. Findings are described by focus area (Fig 3).

**Table 2. Characteristics of included sources.**

| | All sources n = 195 (100.0%) | Bangladesh n = 26 (13.3%) | Bhutan n = 2 (1.0%) | India n = 41 (21.0%) | Indonesia n = 93 (47.7%) | Myanmar n = 10 (5.1%) | Nepal n = 14 (7.2%) | Sri Lanka n = 10 (5.1%) | Thailand n = 4 (2.1%) | Timor Leste n = 3 (1.5%) |
|---|---|---|---|---|---|---|---|---|---|---|
| **Publication Type** | | | | | | | | | | |
| Peer reviewed journal article | 170 (87.2) | 19 (9.7) | 2 (1.0) | 35 (17.9) | 89 (45.6) | 8 (4.1) | 11 (5.6) | 8 (4.1) | 4 (2.1) | 2 (1.0) |
| Conference abstract/paper | 23 (11.8) | 6 (3.1) | 0 (0.0) | 6 (3.1) | 4 (2.1) | 2 (1.0) | 2 (1.0) | 2 (1.0) | 0 (0.0) | 1 (0.5) |
| Abstract (other) | 2 (1.0) | 1 (0.5) | 0 (0.0) | 0 (0.0) | 0 (0.0) | 0 (0.0) | 1 (0.5) | 0 (0.0) | 0 (0.0) | 0 (0.0) |
| **Location of Author Affiliations*** | | | | | | | | | | |
| In country reported on | 168 (86.2) | 18 (9.2) | 0 (0.0) | 36 (18.5) | 86 (4.4) | 6 (3.1) | 6 (3.1) | 9 (4.6) | 3 (1.5) | 3 (1.5) |
| External to country reported on | 98 (50.3) | 24 (12.3) | 2 (1.0) | 21 (10.8) | 26 (13.3) | 9 (4.6) | 12 (6.2) | 7 (3.6) | 2 (1.0) | 3 (1.5) |
| **Methodology** | | | | | | | | | | |
| Quantitative | 94 (48.2) | 9 (4.6) | 1 (0.5) | 20 (10.3) | 52 (26.7) | 3 (1.5) | 5 (2.6) | 5 (2.6) | 1 (0.5) | 1 (0.5) |
| Qualitative | 67 (34.4) | 10 (5.1) | 1 (0.5) | 14 (7.2) | 30 (15.4) | 5 (2.6) | 5 (2.6) | 4 (2.1) | 2 (1.0) | 1 (0.5) |
| Mixed methods | 31 (15.9) | 6 (3.1) | 0 (0.0) | 7 (3.6) | 10 (5.1) | 2 (1.0) | 3 (1.5) | 1 (0.5) | 1 (0.5) | 1 (0.5) |
| Not reported | 3 (1.5) | 1 (0.5) | 0 (0.0) | 0 (0.0) | 1 (0.5) | 0 (0.0) | 1 (0.5) | 0 (0.0) | 0 (0.0) | 0 (0.0) |
| **Focus Area*** | | | | | | | | | | |
| Practice or service delivery | 73 (37.4) | 9 (4.6) | 1 (0.5) | 10 (5.1) | 39 (20.0) | 3 (1.5) | 4 (2.1) | 4 (2.1) | 1 (0.5) | 2 (1.0) |
| Pre-service education | 60 (30.8) | 16 (8.2) | 1 (0.5) | 13 (6.7) | 26 (13.3) | 3 (1.5) | 4 (2.1) | 0 (0.0) | 1 (0.5) | 0 (0.0) |
| In-service education/CPD | 51 (26.2) | 7 (3.6) | 0 (0.0) | 13 (6.7) | 18 (9.2) | 1 (0.5) | 7 (3.6) | 4 (2.1) | 1 (0.5) | 0 (0.0) |
| Workforce management | 46 (23.6) | 3 (1.5) | 0 (0.0) | 12 (6.2) | 18 (9.2) | 5 (2.6) | 3 (1.5) | 2 (1.0) | 1 (0.5) | 2 (1.0) |
| Governance and regulation | 21 (10.8) | 4 (2.1) | 1 (0.5) | 7 (3.6) | 9 (4.6) | 2 (1.0) | 5 (2.6) | 0 (0.0) | 1 (0.5) | 0 (0.0) |
| Leadership | 12 (6.2) | 3 (1.5) | 0 (0.0) | 5 (2.6) | 4 (2.1) | 0 (0.0) | 0 (0.0) | 0 (0.0) | 0 (0.0) | 0 (0.0) |
| **Midwifery Focus*** | | | | | | | | | | |
| Midwives or nurse midwives | 101 (51.8) | 14 (7.2) | 1 (0.5) | 10 (5.1) | 68 (34.9) | 0 (0.0) | 2 (1.0) | 1 (0.5) | 2 (1.0) | 3 (1.5) |
| Auxiliary midwives or auxiliary nurse midwives | 33 (16.9) | 0 (0.0) | 0 (0.0) | 21 (10.8) | 0 (0.0) | 6 (3.1) | 6 (3.1) | 0 (0.0) | 0 (0.0) | 0 (0.0) |
| Students or trainees | 33 (16.9) | 4 (2.1) | 0 (0.0) | 7 (3.6) | 20 (10.3) | 1 (0.5) | 0 (0.0) | 0 (0.0) | 1 (0.5) | 0 (0.0) |
| Midwifery leaders or educators | 26 (13.3) | 10 (5.1) | 0 (0.0) | 5 (2.6) | 9 (4.6) | 0 (0.0) | 2 (1.0) | 0 (0.0) | 0 (0.0) | 0 (0.0) |
| Midwifery profession or education | 14 (7.2) | 5 (2.6) | 1 (0.5) | 3 (1.5) | 4 (2.1) | 3 (1.5) | 5 (2.6) | 0 (0.0) | 1 (0.5) | 0 (0.0) |
| Public health midwives | 9 (4.6) | 0 (0.0) | 0 (0.0) | 0 (0.0) | 0 (0.0) | 0 (0.0) | 0 (0.0) | 9 (4.6) | 0 (0.0) | 0 (0.0) |

*All percentages calculated from a total of 195 sources; totals may not equal 100% as some sources had multiple author affiliations and reported on multiple focus areas, midwifery foci and countries

## Practice or service delivery

Of the 195 sources, 73 (37.4%) reported on an aspect of practice or service delivery. Most were from Indonesia (n = 39, 20.0%), India (n = 10, 5.1%) and Bangladesh (n = 9, 4.6%). Settings included midwife-led models of care, independent practices of midwifery, clinics, community-based services, and hospitals. Areas of clinical practice examined included screening for pre-eclampsia [13], HIV counselling and testing [14], managing anaemia [15, 16], provision of long-acting and permanent methods of contraception [17], antenatal depression knowledge [18], breastfeeding support [19, 20], mental healthcare provision [21], working with women who self-harm [22], and provision of misoprostol [23]. Many studies identified barriers and enablers in the provision of care including a lack of available protocols and/or guidelines to support care

1. **Practice or service delivery**

   This focus area relates to midwives' scope of practice, models of care including midwifery-led continuity of care, safe and supportive practice or service settings, optimised service delivery, and the midwives practicing to the full scope of their competencies.

2. **Pre-service education**

   Education or training programmes and pathways were described for the production of midwives equipped to practice within the International Confederation of Midwives' competencies, or for the production of auxiliary nurse midwives or auxiliary midwives to address workforce shortages.

3. **In-service education or continuing professional development**

   This focus area included continued education and training which enables midwives and auxiliary nurse midwives/midwives to maintain current skills and knowledge. Continued education enabled midwives to advance into leadership, research and education positions.

4. **Workforce management**

   Research mapped to this focus area encompassed the effective distribution of the midwifery workforce. This could include workforce planning, job creation, the midwifery shortage, and recruitment or retention of midwives.

5. **Governance and regulation**

This focus area included research into the governance of midwives' registration and scope of practice, including professional and regulatory bodies, accreditation frameworks, policies, and legislation.

6. **Leadership**

   Research could report on midwifery leadership in management, strategic decision-making, policymaking, and governance. Investments in developing leadership skills and pathways were also described.

**Fig 2. Definition of scoping review focus areas.**

provision [13, 18, 23]. A lack of professional development in clinical care provision was identified as a barrier to quality clinical-based care [19]. Further aspects of practice or service delivery

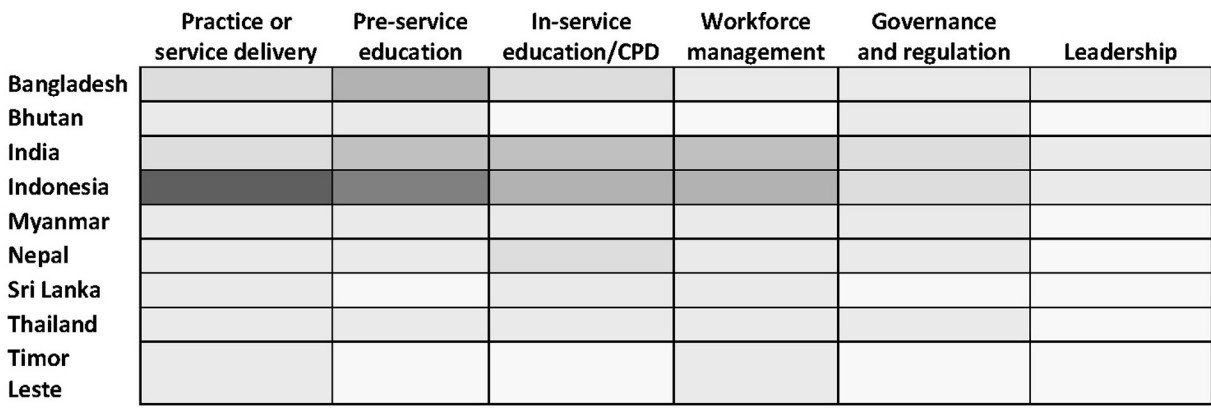

**Fig 3. Heat map of focus area by country.**

included evaluation of midwife-led services or introduction of midwives to maternity services [24–32], service delivery in hard to reach or remote settings [23, 33–38] and the integration of mHealth (i.e. digital health applications) to optimise service delivery [16, 39–48].

Nine sources specifically evaluated the introduction of midwives to maternity services or midwifery-led models of care in Bangladesh [25, 27–29], Indonesia [30, 31], India [32] and Thailand [24]. In response to the introduction of midwives, increases were reported in WHO-recommended birth practices [25], overall service capacity [27, 29], maternal and neonatal health outcomes [30, 32] and women's satisfaction [32]. Two sources pre-emptively explored factors influencing the introduction of nurse-practitioners in midwifery [49] or independent nurse-midwifery practitioners [50] in India. There were examples of partnerships between midwives and community or village health volunteers [13, 24, 51], health cadres [30, 39] and traditional birth attendants [52].

Under-resourcing and workforce shortages were consistent challenges in remote or hard-to-reach settings [33, 35, 37, 53]. Midwives, auxiliary nurse midwives and auxiliary midwives were further challenged by pressure to practice beyond their scope, confidence, personal expectations or training [23, 35, 37]. Midwifery care provision was reported as acceptable in communities [54], however there was a lack of acceptance and community cooperation, and a mismatch of expectations of the role of the midwife [35, 55]. Whilst midwives reported some satisfaction with roles and experiences of quality care provision [56–58], a perceived lack of professional recognition and associated scope was reported as problematic [54, 59]. Key factors to successful integration of midwifery care in remote communities were community trust [35, 53], knowledge of culture and language [37, 53], living in the community [23, 38, 53] and length of service in that community [35, 60]. It was clear that the midwives should be integrated into their communities, and communities be informed of the benefits of midwifery care.

Finally, 11 sources reported on the integration of mHealth phone applications or internet-based systems to optimise practice or service delivery. These were conducted in Indonesia [39–43, 48], India [16, 44], Timor-Leste [45], Myanmar [46] and Sri Lanka [47]. mHealth interventions were used to facilitate communication between midwives and women, including the provision of health information and appointment reminders [40, 42, 45]. They were also designed to improve documentation [43, 47], support auxiliary midwives' clinical decision-making [46], facilitate the targeted provision of services [39], to facilitate referrals between primary health centres and hospitals [41] and to facilitate monitoring of anaemia [16]. The acceptance of mHealth applications by auxiliary nurse midwives [44] and midwives' mHealth competency [48] were also explored more broadly.

## Pre-service education

A total of 60 sources (30.8%) reported on pre-service education from Indonesia (n = 26, 13.3%), Bangladesh (n = 16, 8.2%) and India (n = 13, 6.7%). Twelve reported on the importance of strengthening specific aspects of midwifery education, including the development of legislation [61, 62] and regulation [63], formal midwifery education [64, 65] and professional value [66, 67]. Improved learning environments were recommended to strengthen formal midwifery education. Major gaps and bottlenecks were also identified in pre-service education inhibiting their scale up, quality improvement efforts and reform [63]. Gaps related to infrastructure, resources and capacities for midwifery pre-service education [65]. Strategies to address these included: 1) legislation and regulation; 2) training and education; 3) deployment and utilisation; and 4) professional associations [65, 68].

National accreditation standards were reported in four papers. These included the development of context-specific accreditation assessment tools and a rapid assessment tool for

affirming good practice in midwifery education programming [66]. In Bangladesh, an accreditation process improved communication between midwifery teaching institutions, policymakers and regulatory authorities [69]. A globally validated tool was also developed along with a user guide and handbook [68].

Twenty-seven sources reported on initiatives and processes to improve learning. Examples were a flipped classroom approach [70], case analysis activities, integration of women's voices through storytelling videos of women's childbirth experiences [71], team-based learning [72, 73], augmented reality-based media [74], competency tests, preceptorship, simulation, e-learning [75] and blended learning approaches [76]. Learning experiences in practice included learning in, and about, a continuity of care model [77] and being based in a community midwifery clinic [78]. Another focus was improving confidence of students such as clinical skill development with hands-on clinical practice or interprofessional education. Interventions related to HIV/AIDS knowledge and attitudes [79], postpartum haemorrhage [72] and a mini clinical evaluation exercise [80] were evaluated using a pre-and-post-test approach. Development of the midwifery faculty or teachers was also recognised as being important to improve teaching and learning initiatives [81]. Nine sources reported on faculty or curriculum development including masters' programs [81], international midwifery placements and a mentorship programme [82]. Training for midwife supervisors on the provision of effective feedback was another initiative [78].

## In-service education or continuing professional development

A total of 51 sources reported on in-service education or CPD, primarily in Indonesia (n = 18, 9.2%), India (n = 13, 6.7%), Nepal (n = 7, 3.6%) and Bangladesh (n = 7, 3.6%). In one report, midwives in Indonesia accessed a model of learning, known as the AmmuntuliBija intervention model, which improved knowledge, motivation and abilities to provide antenatal services [83]. Low dose-high frequency simulation training, such as the Helping Babies Breathe Program, showed improvements in midwives' skills and knowledge [84]. Other specific programs related to positive pressure ventilation [85], abortion care [34, 51] and emergency care [86]. Approaches included high- fidelity simulation, facilitated video-guided debriefing and facilitated teamwork activities [87]. Targeted in-service education referred to essential maternal and newborn health-related skills. Other focus areas included partograph use [88], neonatal care [84], use of alternate birth positions [29] and antenatal care [43], HIV [89], family planning [90], mental health [91], ultrasound [92], patient safety [93], family violence [94], digestion health, nutrition and parenting [95], breast cancer [96], mammography [97], disaster preparedness and management [98], electronic health record system [47] and pharmacovigilance [99]. Targeted in-service education was offered for career pathway development, specifically professional education, academic progress, self-development or professional recognition, or with external incentives such as career paths, policies, workplace or professional organisation.

Finally, limitations in post-registration knowledge and skills were identified. These included low levels of knowledge of critical danger signs [100] including safe childbirth practices, exposure to rural communities [56], medical legal issues and ethics [101], folic acid supplementation [102], blood loss assessment [103], intrauterine device counselling [104], vulvovaginal discharge and health education [105].

## Workforce management

Of the included sources, 46 papers (23.6%) explored aspects of workforce management predominantly in Bangladesh (n = 18, 9.2%), India (n = 12, 6.2%) and Myanmar (n = 5, 2.6%).

Common themes included the importance of work/life balance [106–108] supportive work environments [107, 109, 110], clear career paths [49, 60, 111], and ongoing training and supervision [56, 112] to promote professional sustainability. Local cultural acceptance and the social status of the midwife was reported as critical to the effective provision of care [38, 60, 113]. The implementation of local or village midwives was shown to improve health outcomes for whole communities [38, 52].

Challenges included difficulty accessing remote areas [33, 37, 53], and threat to personal safety [98] which led to unwillingness to work in these areas [36, 114]. Korake and Nilima [113] reported that midwives working in rural India experienced issues that effected their safety and wellbeing including, poor accommodation, safety and security threats, gender inequalities, expectation of working long hours, and limited communication and transportation facilities. Burnout and stress were identified as a threat to workforce sustainability [37, 108–110]. Higher rates of burnout were reported to be trending in younger midwife populations and those who did not perceive adequate pay in one Indonesian study [115].

## Governance and regulation

Twenty-one sources from seven countries reported on governance and regulation, primarily Indonesia (n = 9, 4.6%), India (n = 7, 3.6%) and Nepal (n = 5, 2.6%). The primary priority for midwifery governance was identified as access to quality data to advise and improve regulation, education, safety and quality monitoring and service provision reported in seven papers [49, 116–120]. Midwifery representation on governance committees was identified as critical to advancing midwifery and ensuring that key elements of the profession are represented in regulatory and legal forums in three studies [49, 63, 119].

The variance in approaches to legislation and midwifery regulation especially in relation to nursing and other auxiliary roles was a barrier to advancing midwifery [49, 63, 112, 121, 122]. A lack of clarity or absence of legislation supporting midwifery practice, particularly in community settings was identified as a barrier to the provision of primary care [123] with midwives fearful of working outside legal frameworks [17, 101, 124]. Consistent regulation and accreditation of midwifery education was reported as still emerging, with professional identity and progress hindered until this is achieved [49, 61, 62, 65, 66, 112, 116, 117, 125].

## Leadership

Finally, twelve sources (6.2%) explicitly reported on aspects of strengthening leadership in India (n = 5, 2.6%), Indonesia (n = 4, 2.1%) and Bangladesh (n = 3, 1.5%). Three sources reported on the effect of democratic leadership styles implemented by midwife coordinators on village midwives' performance in Indonesia [114, 126, 127]. Most reported on midwifery leadership in India at a national level [49, 63, 111] and in the context of a mobile nurse-midwifery mentor programme to improve maternal and neonatal emergency skills [87, 128]. International mentorship to build capacity amongst midwifery faculty in Bangladesh to deliver a national midwifery diploma, supervise midwifery students and mentor colleagues was reported on [82, 125]. One source from Bangladesh reported on the positive effect of midwifery mentorship on the acceptance of midwives into pre-existing hospital settings, midwives' pride and quality of clinical practices [26].

Midwifery leadership was identified as a strategy to grow midwifery as a profession, and address misconceptions about roles, expertise and capacity [49]. The need for leaders to prioritise the visibility of the identity of the midwifery profession distinct from nursing at strategic levels of national government, through strengthened education, regulation and legislation was highlighted [49, 63, 82, 87, 111, 114, 125, 126]. Cultural endorsement and investment in

midwifery were critical to operationalising midwifery leadership; mentorship of emerging midwifery leaders across education, clinical care, administration, regulation, research and governance was identified as an area of urgent need in three studies [63, 111, 125]. Advocacy for scope fulfillment was heavily reliant on leaders paving the way and demonstrating the capacity of midwives to work in a variety of settings and across the life continuum [63, 87, 111, 114, 126].

## Discussion

The aim of this review was to identify what areas of midwifery have been explored in the last 10 years in the WHO SEARO region and to identify future research priorities to advance and strengthen midwifery. In total, 195 papers were analysed. This is more papers than we expected highlighting the considerable research that has been undertaken across this region in the last decade to strengthen midwifery.

Some countries have undertaken an extensive amount of research, for example, Indonesia. This is not unexpected given the size of the midwifery profession and the commitment to strengthen midwifery by the Government of Indonesia and the work of the midwifery society. Countries like Bangladesh have been transitioning to midwifery as a profession in the last decade and are implementing midwifery models of care and midwife-led units. It is encouraging to see research on those initiatives in this review. Other countries have had less attention, for example, Sri Lanka, Bhutan, Thailand and Timor-Leste. These are potentially explainable due to the size of the profession in some of these countries and the focus on nursing rather than midwifery in some. We could not find any research from DPR Korea or the Maldives.

As more countries choose to transition to midwifery-led care and midwifery models of care [129, 130], research needs to be undertaken to better understand the barriers and enablers to this transition and to be able to put in place strategies to support the implementation. Research to understand the effectiveness and implementation of midwifery-led care and midwifery models of care is important to ensure that these approaches are well evaluated and lessons can be learned for other countries and regions. Finally, research on the return on investment in midwifery education, regulation deployment and retention would be beneficial to support countries in making strategic decisions around investments. Any research on midwifery education and the capacity to meet international standards needs to also consider the local context and culture and the barriers or enablers to ensuring quality midwifery education.

An identified gap was a need to better understand of how to build and sustain effective midwifery leadership. The importance of leadership is mentioned in almost all global reports that aim to strengthen midwifery leadership but the 'how' is often missing. An investment in midwifery leadership development across the various sectors (education, service delivery, policy, regulation) is an area of critical need that will support sustainable approaches to strengthening midwifery within the region and globally. Strengthening midwifery leadership and encouraging equality through representation of midwifery governance issues by midwifery leaders is critical to advancing the profession and should be prioritised.

Limitations of the review include only peer reviewed research was included in the review. It is possible that non-government organisations or United Nations and other agencies may have undertaken relevant research but have not published these findings. Ensuring that research on midwifery has an opportunity to be published in the peer reviewed literature is important for future similar analyses. In addition, research conducted by doctoral or masters' students, or published in local professional and in local languages outlets, was not readily accessible through the selected databases. The development of an internationally accessible database may facilitate exchange of research and knowledge between countries in the region, to learn from one another. Sources were not excluded by language or publication. However, only sources

published in English or Indonesian were captured in this search strategy, suggesting that local databases were not captured.

## Conclusion

This review has identified the areas of midwifery that have been explored in the last 10 years in the WHO SEARO region and has identified future research priorities to advance and strengthen midwifery. The number of papers highlighted the considerable research that has been undertaken across the region in the last decade to strengthen midwifery. More research on the various aspects to strengthen midwifery in the next decade would support strategic and targeted planning for continued growth.

## Supporting information

**S1 Table. Search terms used.**
(DOCX)

**S2 Table. All included sources.**
(DOCX)

**S1 Checklist. Preferred Reporting Items for Systematic reviews and Meta-Analyses extension for Scoping Reviews (PRISMA-ScR) checklist.**
(DOCX)

## Acknowledgments

Thank you to delegates at the WHO's South East Asian Regional Office (SEARO) Midwifery Regional meeting to review progress and strengthen midwifery programme in South-East Asia Region, Dhaka, Bangladesh, 13–15 September 2022 who provided additional materials to assist with the conduct of the review. Thank you also to Vanessa Varis, Faculty Librarian at the Faculty of Health Sciences and Vice Chancellory, Curtin University in Australia for her feedback on development of the search strategy.

## Author Contributions

**Conceptualization:** Georgia Griffin, Zoe Bradfield, Rachel Smith, Ai Tanimizu, Neena Raina, Caroline S. E. Homer.

**Data curation:** Georgia Griffin, Zoe Bradfield, Rachel Smith, Caroline S. E. Homer.

**Formal analysis:** Georgia Griffin, Zoe Bradfield, Kyu Kyu Than, Rachel Smith, Caroline S. E. Homer.

**Funding acquisition:** Ai Tanimizu, Neena Raina, Caroline S. E. Homer.

**Methodology:** Zoe Bradfield, Ai Tanimizu, Caroline S. E. Homer.

**Project administration:** Caroline S. E. Homer.

**Resources:** Neena Raina.

**Supervision:** Caroline S. E. Homer.

**Writing – original draft:** Georgia Griffin, Zoe Bradfield, Kyu Kyu Than, Rachel Smith, Ai Tanimizu, Caroline S. E. Homer.

**Writing – review & editing:** Georgia Griffin, Zoe Bradfield, Kyu Kyu Than, Rachel Smith, Ai Tanimizu, Neena Raina, Caroline S. E. Homer.

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
