## [Decision Letter · Decision Letter 0]

17 Oct 2023

PONE-D-23-28039Strengthening midwifery in the South-East Asian region: A scoping review of midwifery-related researchPLOS ONE

Dear Dr. Homer,

Thank you for submitting your manuscript to PLOS ONE. After careful consideration, we feel that it has merit but does not fully meet PLOS ONE’s publication criteria as it currently stands. Therefore, we invite you to submit a revised version of the manuscript that addresses the points raised during the review process.

We look forward to receiving your revised manuscript.

Kind regards,

Ivan Sarmiento

Academic Editor

PLOS ONE

Journal Requirements:

3. Please expand the acronym “WHO SEARO” (as indicated in your financial disclosure) so that it states the name of your funders in full.

 "The WHO SEARO office commissioned the study and were part of the analysis and writing team."

6. Please ensure that you refer to Figure 3 in your text as, if accepted, production will need this reference to link the reader to the figure.

**Additional Editor Comments:**

Unfortunately, it has been challenging to find reviewers for your paper. As a result, I have taken the responsibility of completing the second review myself to expedite the process. 

 Even though I have requested major revisions, this is only to provide you with more time to address the comments. Please be consistent with the use of active voice throughout the paper.  I suggest you acknowledge the limitations associated with excluding sources reporting on midwifery programs without peer review. Many NGO and official institutions might have published this type of report.  The first ten lines of the data charting section belong to the results as well as the reference to supplementary file 2. I suggest discussing the implications of your findings for how to approach midwifery programs in relation to following international standards or contextualising the programs in local conditions. There is an important movement for defining and imposing international standards which might not always align with the diversity that you identified in your review. It is also possible to discuss the significance of culturally safe approaches to midwifery practice. As you mentioned towards the end of page 10, understanding the local context and establishing a good relationship with communities is crucial. Even more, Indigenous groups in Australia and New Zealand have expressed their concerns about programs that disrupt local and indigenous communities, for example, with initiatives trying to replace traditional midwives.

Reviewers' comments:

Reviewer's Responses to Questions

**Comments to the Author**

1. Is the manuscript technically sound, and do the data support the conclusions?

Reviewer #1: Yes

2. Has the statistical analysis been performed appropriately and rigorously? 

Reviewer #1: N/A

3. Have the authors made all data underlying the findings in their manuscript fully available?

Reviewer #1: Yes

4. Is the manuscript presented in an intelligible fashion and written in standard English?

Reviewer #1: Yes

5. Review Comments to the Author

Reviewer #1: General comment

I want to start by congratulating the authors on this high-quality manuscript. This scoping review addresses an important issue and offers a broad perspective on midwifery care. While the review focuses on the South-East Asia Region, its findings are also applicable to other Regions. The methodology is exceptionally well-conducted and could serve as an example for other regions to follow. The manuscript is well-written and contains relevant messages that can contribute to the global advancement of midwifery. I eagerly anticipate the publication of this manuscript and will promote it as an exemplary resource for my master’s and PhD students.

Detailed comments

- Pdf pg 10: insert a space between outcomes and [1] (“Thus, empowering midwives by prioritising their education and training are crucial to achieving sustainable improvements in sexual, reproductive, maternal, neonatal and adolescent health outcomes[1].”)

- Pdf pg 11: personally, I would prefer to see the 5 phases better reflected in the titles of the paragraphs in the method section

- Pdf pg 12: inconsistency in writing eleven / 11 countries

- Pdf pg 12: inconsistency in writing South-East Asia region/ Region

- Pdf pg 13: insert a space between possible and [11] (“Study quality was not appraised, as scoping reviews include a broad range of evidence sources and this is not usually possible[11].”)

- Pdf pg 14: correction CPD = continuous professional development instead of continuous professional delivery (CPD)

- Pdf pg 16: Is it possible that we miss a word in the following sentence? “It was clear that should be integrated into their communities, and communities be informed of the benefits of midwifery care.

6. PLOS authors have the option to publish the peer review history of their article (what does this mean?). If published, this will include your full peer review and any attached files.

Reviewer #1: **Yes: **Prof. dr. Mieke Embo

---

## [Author Response · Author response to Decision Letter 0]

26 Oct 2023

Thank you for the opportunity to respond to these very helpful reviews. We really appreciate the timeliness of this review. We have addressed each comment in turn and have track changes in the manuscript. 

3. Please expand the acronym “WHO SEARO” (as indicated in your financial disclosure) so that it states the name of your funders in full.

Thank you. This is corrected. 

 "The WHO SEARO office commissioned the study and were part of the analysis and writing team."

Please state what role the funders took in the study. If the funders had no role, please state: ""The funders had no role in study design, data collection and analysis, decision to publish, or preparation of the manuscript."" If this statement is not correct you must amend it as needed.

The WHO SEARO office were involved in the study design, interpretation and decision to publish. They are included as co-authors. We have added this in the Cover Letter and the manuscript. 

The data are all extracted from published papers into an Excel spreadsheet. This is available in Supplementary Table 2. 

6. Please ensure that you refer to Figure 3 in your text as, if accepted, production will need this reference to link the reader to the figure.

This is now included. 

This is now included. 

Additional Editor Comments:

Please be consistent with the use of active voice throughout the paper. 

 We have altered the text in a number of places to address this.

I suggest you acknowledge the limitations associated with excluding sources reporting on midwifery programs without peer review. Many NGO and official institutions might have published this type of report. 

Thank you. We have added that.

The first ten lines of the data charting section belong to the results as well as the reference to supplementary file 2.

This is moved and a small section on data charting is now included.

I suggest discussing the implications of your findings for how to approach midwifery programs in relation to following international standards or contextualising the programs in local conditions. There is an important movement for defining and imposing international standards which might not always align with the diversity that you identified in your review.

Thank you. I have added this.

It is also possible to discuss the significance of culturally safe approaches to midwifery practice. As you mentioned towards the end of page 10, understanding the local context and establishing a good relationship with communities is crucial. Even more, Indigenous groups in Australia and New Zealand have expressed their concerns about programs that disrupt local and indigenous communities, for example, with initiatives trying to replace traditional midwives.

Thank you. I have added this.

Reviewer #1: General comment

I want to start by congratulating the authors on this high-quality manuscript. This scoping review addresses an important issue and offers a broad perspective on midwifery care. While the review focuses on the South-East Asia Region, its findings are also applicable to other Regions. The methodology is exceptionally well-conducted and could serve as an example for other regions to follow. The manuscript is well-written and contains relevant messages that can contribute to the global advancement of midwifery. I eagerly anticipate the publication of this manuscript and will promote it as an exemplary resource for my master’s and PhD students.

Thank you. We appreciate this positive comment and hope it will be a useful paper when published. 

Detailed comments

- Pdf pg 10: insert a space between outcomes and [1] (“Thus, empowering midwives by prioritising their education and training are crucial to achieving sustainable improvements in sexual, reproductive, maternal, neonatal and adolescent health outcomes[1].”)

Corrected.

- Pdf pg 11: personally, I would prefer to see the 5 phases better reflected in the titles of the paragraphs in the method section

Thank you – this is a good suggestion. We have done this now.

- Pdf pg 12: inconsistency in writing eleven / 11 countries

Thanks for picking this up – I have used the numeral. 

- Pdf pg 12: inconsistency in writing South-East Asia region/ Region

Thank you. I have corrected this.

- Pdf pg 13: insert a space between possible and [11] (“Study quality was not appraised, as scoping reviews include a broad range of evidence sources and this is not usually possible[11].”)

Thank you. I have corrected this.

- Pdf pg 14: correction CPD = continuous professional development instead of continuous professional delivery (CPD)

Thank you. I have corrected this.

- Pdf pg 16: Is it possible that we miss a word in the following sentence? “It was clear that should be integrated into their communities, and communities be informed of the benefits of midwifery care.

Thank you. I have corrected this.

---

## [Editor Report · Decision Letter 1]

30 Oct 2023

Strengthening midwifery in the South-East Asian region: A scoping review of midwifery-related research

PONE-D-23-28039R1

Dear Dr. Homer,

We’re pleased to inform you that your manuscript has been judged scientifically suitable for publication and will be formally accepted for publication once it meets all outstanding technical requirements.

Kind regards,

Ivan Sarmiento

Academic Editor

PLOS ONE
---

## [Editor Report · Acceptance letter]

17 Nov 2023

PONE-D-23-28039R1 

Strengthening midwifery in the South-East Asian region: A scoping review of midwifery-related research 

Dear Dr. Homer:

I'm pleased to inform you that your manuscript has been deemed suitable for publication in PLOS ONE. Congratulations! Your manuscript is now with our production department. 

Kind regards, 

on behalf of

Dr. Ivan Sarmiento 

Academic Editor

PLOS ONE